# The Molecular Regulatory Pathways and Metabolic Adaptation in the Seed Germination and Early Seedling Growth of Rice in Response to Low O_2_ Stress

**DOI:** 10.3390/plants9101363

**Published:** 2020-10-14

**Authors:** Mingqing Ma, Weijian Cen, Rongbai Li, Shaokui Wang, Jijing Luo

**Affiliations:** 1College of Life Science and Technology (State Key Laboratory for Conservation and Utilization of Subtropical Agro-Bioresources), Guangxi University, Nanning 530004, China; mqma@st.gxu.edu.cn (M.M.); cweijian@gxu.edu.cn (W.C.); lirongbai@126.com (R.L.); 2Agriculture College, Guangxi University, Nanning 530004, China; 3Agriculture College, South China Agricultural University, Guangzhou 510642, China; shaokuiwang@scau.edu.cn

**Keywords:** submergence, direct seeding, anaerobic germination, low O_2_ stress, regulatory mechanism, metabolic adaptation

## Abstract

As sessile organisms, flooding/submergence is one of the major abiotic stresses for higher plants, with deleterious effects on their growth and survival. Therefore, flooding/submergence is a large challenge for agriculture in lowland areas worldwide. Long-term flooding/submergence can cause severe hypoxia stress to crop plants and can result in substantial yield loss. Rice has evolved distinct adaptive strategies in response to low oxygen (O_2_) stress caused by flooding/submergence circumstances. Recently, direct seeding practice has been increasing in popularity due to its advantages of reducing cultivation cost and labor. However, establishment and growth of the seedlings from seed germination under the submergence condition are large obstacles for rice in direct seeding practice. The physiological and molecular regulatory mechanisms underlying tolerant and sensitive phenotypes in rice have been extensively investigated. Here, this review focuses on the progress of recent advances in the studies of the molecular mechanisms and metabolic adaptions underlying anaerobic germination (AG) and coleoptile elongation. Further, we highlight the prospect of introducing quantitative trait loci (QTL) for AG into rice mega varieties to ensure the compatibility of flooding/submergence tolerance traits and yield stability, thereby advancing the direct seeding practice and facilitating future breeding improvement.

## 1. Introduction

Rice (*Oryza sativa* L.) is a staple food crop feeding more than half of the world’s population [1]. The traditional rice production system in Asian countries commonly involves transplanting seedlings from the nursery into a paddy field. This production pattern is labor-, water-, and energy-consuming and is becoming less profitable [2]. Therefore, in recent years, direct seeding has been receiving much attention worldwide, especially in Asian countries, and farmers have shifted to direct seeding of rice due to its low cost and labor-saving strategy [2,3,4]. There are three major methods for rice direct seeding: dry seeding (sowing dry seeds into dry soil), wet seeding (sowing pregerminated seeds on wet puddled soils), and water seeding (sowing seeds into standing water) [5]. However, poor crop stand establishment has become a key obstacle that prevents the subsequent growth, development, and yield in the direct seeding of rice [5]. In particular, water seeding severely affects the establishment of rice seedlings due to low water oxygen under the submergence condition.

For the early emerging seedlings, submergence reduces air diffusion, which limits O_2_ and carbon dioxide (CO_2_) availability in the submerged tissues, resulting in a hypoxic (<21% O_2_) environment, with an O_2_ concentration below that under normoxic conditions [6,7,8,9]. At low O_2_ supply (hypoxia) conditions, O_2_ concentration can affect the respiration in plants, and respiration can still active although at a strong reduced rate when O_2_ concentration is below the critical oxygen pressure for respiration (COPR), which is the lowest oxygen partial pressure to support maximum respiration rate [10,11]. Under O_2_ deficiency conditions, the accumulation of several phytotoxic substances such as reduced iron (Fe^2+^), manganese (Mn^2+^), hydrogen sulfide (H_2_S), oxygen radicals, and the products of fermentation, cause severe damage to plants. The ability of rice seeds to avoid the damages caused by these toxins is helpful to prevent further injury [4,12]. Although most modern rice varieties have low ability to germinate under water, rice has developed various adaptive mechanisms morphologically and physiologically in order to adapt to a wide range of hydrological environments. There are genetic variations exist among rice varieties, and these variations lay the base for the possibility of studying the molecular regulatory mechanism of AG-related traits and facilitate breeding rice varieties that are suitable for direct seeding systems [4]. In direct seeding practice, varieties that are suitable for water seeding have characteristics of high anaerobic germination tolerance, early vigorous seedling growth, fast root growth, early tillering, and lodging resistance [13]. Therefore, the breeding of rice varieties that are capable of surviving and accommodating submergence conditions during germination and early growth stages improves the success of direct-seeded rice [14]. Thus, prior to breeding applications, it is of great importance to isolate related genes for AG tolerance from tolerant genotypes and to understand the genetic basis and the molecular regulatory mechanisms underlying anaerobic germination and submerged seedling growth.

Despite the sophisticated molecular mechanisms involved in the regulation of the anaerobic germination and seedling growth in rice, considerable progress on understanding the genetic, molecular and physiological basis of rice in response to submergence stress at the germination stage, has been achieved in the last few decades [15,16,17,18,19,20,21,22,23]. Here, we mainly focus on reviewing the molecular genetic mechanisms that regulate the metabolic adaptation of rice to anaerobic germination stress; we could not cover all the related advances due to space limitations. The advances reviewed here are crucial for understanding the fundamental mechanisms and for the breeding design in direct seeding practice. Last but not least, we proposed future research prospects and premised the issues raised that remain for further investigation.

## 2. Strategies are Adopted in Rice under Submerged Germination

Anaerobic germination (AG) tolerance is an important trait required for rice to successfully germinate in a direct seeding system. Rice has the ability to perceive low O_2_ stress and regulate germination and early seedling growth under hypoxic conditions [24,25]. There are two distinct adaptive responses or survival strategies for coping with submergence stress in plants: one is low O_2_ escape strategy (LOES), and the other is low O_2_ quiescence strategy (LOQS) [26,27,28,29]. In the case of the established rice plants, the two adaptive strategies for flooding tolerance are *SUBMERGENCE 1A* (*SUB1A*)-dependent quiescence strategy and *SNORKEL 1* and *SNORKEL 2* (*SK1/2*)-dependent escape strategy [30]. Overexpression of *SUB1A* stimulated the expression of *alcohol dehydrogenase* (*ADH*) genes for ethanolic fermentation and conferred submergence tolerance at the vegetative stage in rice by up-regulating the expression of *slender rice-1* (*SLR1*) and *SLR1 like-1*(*SLRL1*), two negative regulators for gibberellin (GA) signaling, thereby inhibiting rice elongation, reducing the carbohydrate consumption, and exhibiting a strong submergence tolerance in rice plants. The rice varieties with *SUB1A* gene could survive up to two weeks of the complete submergence [31,32,33]. For *SNORKEL 1* and *SNORKEL 2* (*SK1/2*)-dependent escape strategy, *SK1/2* promotes the internode elongation via the stimulation of GA biosynthesis in deep-water rice under flooding stress, and thereby enabling rice grows upward to the water surface for air exchange [34].

During germination and early seeding stages of rice, the LOES is characterized by promoting the elongation of the mesocotyl and coleoptile to allow plants to reach the water surface to increase air exchange between the aerial and submerged tissues. In rice, varieties that have adopted LOES in response to the submergence stress during germination are considered stress-tolerant [24]. Conversely, LOQS is a strategy that constrains the elongation of the mesocotyl and coleoptile and preserves energy for prolonged submergence stress. The varieties that exhibit slow coleoptile growth under low O_2_ conditions are sensitive to submergence stress, thereby impairing the subsequent growth and development of the seedlings (Figure 1). Therefore, coleoptile elongation is a main index for selection of anerobic germination-tolerant rice varieties.

Previous studies have revealed that cell elongation contributes to the rapid increase in the length of the coleoptile. During the early stage of germination, cell division is activated to increase cell numbers for coleoptile growth, and the subsequent cell expansion contributes to elongation of the coleoptile under anerobic conditions [25,35,36]. Some specific expansins were reported to contribute to coleoptile elongation in rice under anaerobic conditions [25,37,38]; for example, *EXPA2*, *EXPA4*, *EXPA1*, *EXPB11*, and *EXPB17* have been reported to be highly expressed in the coleoptiles under submergence stress [36,37,39]. To adapt to the submergence environment during the germination and early stages of growth, rice plants have evolved specific regulatory mechanisms to modulate metabolic shifts in response to low O_2_ stress [40,41], which will be described in more detail in the following sections.

## 3. Metabolic Adaption to Anaerobic Germination in Rice

In the submerged germination of rice, aerobic respiration is severely inhibited due to water O_2_ deficiency, leading to differential expression of the genes that contribute to activating the essential mechanisms and, in turn, to shifting the metabolic processes to anerobic respiration, which is related to energy production and utilization in the germinating seeds [42]. These alternative pathways to aerobic respiration are known as fermentation, including alcoholic fermentation, lactic acid fermentation, and the alanine pathway [43,44,45]. Meanwhile, tricarboxylic acid (TCA) cycle and oxidative phosphorylation processes are inhibited, and the production of ATP is shifted from electron transport chains (ETCs) to glycolysis and ethanol fermentation. Consequently, the net energy yield from anaerobic fermentation (2 mol ATP per mol of glucose) is lower than that from aerobic respiration (30–32 mol ATP per mol of glucose) (Figure 2). Although limited amounts of energy are produced from ethanol fermentation, it is extremely important for the germination and seedling growth under submergence condition. The energy is allocated to critical bioprocesses, especially protein synthesis in the cells of the coleoptile, contributing to submergence tolerance and success in the establishment of seedlings [46]. Likewise, the reoxidization of NADH to NAD^+^ in anerobic fermentation is also required for continuous glycolysis [6,47]. A recent report suggested that the AG-tolerant genotypes of rice adopt a strategy that strongly increases starch degradation in the endosperm into metabolizable sugars, thereby supplying substrates for subsequent glycolysis and alcohol fermentation to produce the energy required for enhancing the germination vitality and for the rapid outgrowth of the coleoptile [15].

In the starch metabolic pathways, α-amylase (αAmy) (1,4-α-d-glucan maltohydrolase) is one of the most abundant hydrolases in rice and is involved in mobilization of the stored starch [48,49]; it catalyzes the hydrolysis of α-1,4-glucosidic bonds of starch, yielding α-glucose, and α-maltose [50]. In rice, a total of 10 distinct *αAmy* genes are classified into three subfamilies: subfamilies RAmy1 (A, B, and C), RAmy2A, and RAmy3 (A, B, C, D, E, and F). Most of the *αAmy* genes belong to the RAmy3 subfamily [51,52], in which only one gene (LOC_Os08g36910) is induced by sugar starvation and O_2_ deficiency [53]. Thus, RAmy3D is considered an important enzyme in the fermentative metabolism pathway for metabolic shift regulation and in the production of energy in response to submergence stress [19] (Table 1). Regarding the alcoholic fermentation process, ADH, pyruvate decarboxylase (PDC), and aldehyde dehydrogenase (ALDH) are key enzymes that are involved in the reduction of pyruvate into alcohol. ADH catalyzes the rate-limiting step of ethanol metabolism and is considered to be essential for carbohydrate metabolism that is critical for the germination and growth of rice under low O_2_ stress [54,55]. It has been reported that a mutation in rice *ADH1* leads to a reduction in the level of ADH protein, and thus compromises the tolerance to anaerobic stress of the *adh*-null mutant, suggesting that a functional ADH protein is required for enhancing tolerance to submergence during rice germination [20,56] (Figure 2). Moreover, pyruvate is a final product of glycolysis, which is converted to lactate by lactate dehydrogenase (LDH). The reduction in cytoplasmic pH inhibits the activity of LDH and activates the activity of PDC, leading to the reduction of intracellular lactate concentration. Therefore, the regulation of cytoplasmic pH, enables rice to avoid acidosis and improves the survival under low O_2_ conditions [4,25,40]. Therefore, enhancement of the activity of major enzymes for either starch hydrolysis or ethanol synthesis is in favor of promoting the elongation of rice coleoptile under low O_2_ stress [24,39,54,57].

Upon imbibition of cereal grains, the rapid consumption of soluble carbohydrates during the germination and seedling growth stages causes sugar starvation and the activation of αAmy biosynthesis in the scutellum, which is mediated by the CIPK15-SnRK1A-MYBS1 signaling pathway [61]. Among major cereals, only rice produces a complete set of enzymes to degrade starch under AG [19]. With a higher αAmy activity for the degradation of starch to produce energy in germinating seeds, rice promotes coleoptile elongation for surviving the anerobic stress [15,19,53].

## 4. The Molecular Regulatory Pathway of Anaerobic Germination in Rice

As mentioned above, rice submergence-tolerant varieties adopt LOES in response to low O_2_ stress [24]. In submerged germination, O_2_ deficiency and sugar starvation are the key signals for rice responsiveness. O_2_ is one of the key components for the important biochemical reactions in rice cells. When rice germinates under submergence conditions, it senses hypoxic stress and induces the responsive regulatory pathways to adapt to the adverse environment. Calcium (Ca^2+^) signal, acting as a secondary messenger, is involved in multiple plant abiotic/biotic stress responsive signaling pathways [62]. The calcineurin B-like (CBL) proteins, a family of Ca^2+^-binding proteins and one of the major calcium ion sensors in plants, interact with CBL protein-interacting protein kinases (CIPKs) to form the CBL–CIPK complex, thereby decoding Ca^2+^ signaling in response to various abiotic stresses [63,64,65]. CIPKs is a group of plant-specific Ser/Thr protein kinases that contain an N-terminal kinase catalytic domain and a C-terminal regulatory NAF (Asn-Ala-Phe) domain. The NAF domain of CIPKs is sufficient for interaction with CBL protein and leads to the activation of the CIPKs [66].

Recent studies have revealed that the CBL/*CIPK15-SnRK1A-MYBS1*-mediated sugar-signaling cascade plays an important role in low-O_2_ tolerance during the seed germination stage of rice [15,17]. CIPK15 is essential for regulating the expression of genes encoding αAmy and ADH [15] (Figure 2). CBL4 has been reported to be induced by low-oxygen conditions and strongly interacts with the CIPK15 protein. The silence of *CBL4* results in downregulation of the expression of *αAmy3* (*RAmy3 D*). These findings suggest that the CBL4/CIPK15 complex plays a critical role in the sugar starvation signaling pathway and may contribute to the germination of rice under the hypoxia condition [21,67]. OsCBL10, another member of the rice CBL family, has been reported to be involved in flooding stress response during rice germination by comparing flooding-tolerant (Low88) with flooding-intolerant (Up221) cultivars. The natural variation in the *OsCBL10* promoter is associated with the tolerance of anaerobic germination and seedling growth among rice subspecies [16]; according to the sequence variation, its promoter can be characterized into two types, including a flooding-tolerant type (T-type) that only exists in *japonica* lowland cultivars and a flooding-intolerant type (I-type) that exists in *japonica* upland-, and *indica* upland/lowland-, cultivars. The flooding-intolerant cultivars that contain an I-type promoter can downregulate CIPK15 protein accumulation by inducing the expression of *OsCBL10*. In contrast, the cultivars containing a T-type promoter have downregulated *OsCBL10* transcription, higher αAmy activity, and increased CIPK15 protein accumulation [16]. Furthermore, however, the *OsCBL10*-overexpressing line is more sensitive to flooding stress than the wild type, suggesting that *CBL10* negatively regulates the *CIPK15-SnRK1A-MYBS1*-mediated pathway [16].

Rice sucrose nonfermenting-1 (SNF1) homolog SNF1-related protein kinase-1 (SnRK1) is encoded by *SnRK1A* and *SnRK1B*, and SnRK1A is structurally and functionally analogous to SNF1 in yeast and AMP-activated protein kinase (AMPK) in mammals [17,68]. SnRK1A, SNF1, and AMPK are considered as metabolic sensors for monitoring cellular carbohydrate status and/or AMP/ATP levels, play an important role in the regulation of carbon and nitrogen metabolism, and are essential for the proper growth and development of higher eukaryotes [68,69,70]. In the downstream of CBL/CIPK15 complex, SnRK1A is involved in sugar starvation and O_2_ deficiency signaling pathways and acts as a conserved energy and stress sensor under submergence condition [15]. Additionally, SnRK1A-interacting negative regulators (SKINs) physically interact with SnRK1A and negatively regulate its activity to modulate the nutrient starvation signaling and its downstream pathways, including inhibition of the expression of *MYBS1* and *αAmy 3*, consequently resulting in compromising anaerobic germination and seedling growth under submergence stress in rice [22] (Figure 2).

Myeloblastosis sugar response complex 1 (MYBS1) and MYBGA are two crucial transcription factors that can integrate diverse nutrient starvation and GA signaling pathways in downstream of SnRK1A in rice [71]. Under sugar-depleted conditions, MYBS1 specifically binds to the TATCCA *cis*-acting element (the TA-box) that is located on the sugar response complex (SRC) of the *αAmy* promoter, and consequently elevates the activity of αAmy for hydrolysis of starch in the endosperm [17,72,73]. SnRK1A mediates the interaction between MYBS1 and *αAmy 3* SRC, and thus regulates the seed germination and seedling growth of rice [17] (Figure 2). In addition, the phytohormone GA controls diverse aspects of plant growth and development, including seed germination and stem elongation [74]. Endogenous GA is synthesized in embryo and is then transported to the aleurone cells to promote the secretion of αAmy [75,76,77]. Furthermore, MYBGA is a GA-inducible MYB transcription factor that is expressed in cereal aleurone cells [78]. With GA induction, MYBGA upregulates the expression of *αAmy* by binding to the GA-responsive element (GARE) of the *αAmy 3* promoter [75,79,80]. The formation of a stable bipartite MYB (MYBS1-MYBGA)-DNA (GARE-TA box) complex in the cytoplasm coordinates the expression of *αAmy* genes [71]. A recent study revealed that the transcription factor MYBS2 competes with MYBS1 for the same *cis*-element of the *αAmy* promoter and represses *αAmy* expression (Figure 2). Via promoting or inhibiting the expression level of *αAmy*, the competition between MYBS1 and MYBS2 can regulate and maintain sugar within an appropriate range during germination and early seedling growth under sugar starvation stress [23].

Moreover, trehalose-6-phosphate (T6P) metabolism is modulated in response to submergence stress. T6P phosphatase catalyzes the conversion of T6P to trehalose and changes the T6P:sucrose ratio [18,81]. The elevated level of T6P represses the activity of SnRK1A [82] (Figure 2). Therefore, T6P, is considered a sensor of sucrose effectiveness under low oxygen stress and is critical for allocating carbon from source to sink tissues and for maintaining sucrose homeostasis, which influences many metabolic and developmental processes in plants [83,84].

In addition, phytohormones also play an important role in rice under submergence. Ethylene has a regulatory role in the upstream of both the *SUB1A*-dependent quiescence strategy and *SK1/2*-dependent escape strategy [28,85]. GA plays negative roles in *SUB1A*-dependent pathway mediating submergence tolerance and plays positive roles in the *SK1/2*-dependent pathway, triggering rapid internode elongation. GA also promote the elongation of rice coleoptile in the *CIPK15*-dependent pathways under submergence [15,28,40,86].

Thus, *CIPK15* is induced by sugar starvation/low O_2_ signals caused by submergence and then activates the accumulation of SnRK1A protein. The accumulation of SnRK1A triggers a SnRK1A-dependent signaling cascade in response submergence stress. The downstream *SnRK1A-MYBS1-αAmy 3* sugar signaling cascade promotes starch degradation and sugar mobilization, indicating that the *CIPK15-SnRK1A-MYBS1*-mediated sugar sensing pathway plays a key role in the submerged germination of rice seeds [15,17].

## 5. Identification of QTLs/Genes for AG Tolerance

Many QTLs and candidate genes for AG tolerance have been reported in recent years. Coleoptile elongation and seedling survival rate under submerged conditions have been widely used as major indicator traits for tolerance phenotyping in QTL mapping and genome-wide association study (GWAS) approach (Table 2). Among these, five QTLs associated with tolerance of submergence during germination were mapped on chromosomes 1 (*qAG-1-2*), 3 (*qAG-3-1*), 7(*qAG-7-2*), and 9 (*qAG-9-1* and *qAG-9-2*), respectively [87]; four QTLs (*qAG7.1*, *qAG7.2*, *qAG7.3*, and *qAG3*), which derived from *aus* landrace Kharsu 80A for AG tolerance were identified in a recent study [88]. Of which, only one QTL (*qAG-9-2*) has been cloned. *OsTPP7*, a gene encoding T6P phosphatase, was determined to be a target of the QTL *qAG-9-2*. OsTPP7 is involved in the regulation of sugar signaling and in the linking of trehalose metabolism to starch mobilization. The modulation of the local T6P: sucrose ratio by OsTPP7 promotes the flux of sugar from the source (endosperm reserve) toward the sink (embryo axis–coleoptile growth). These processes are partly mediated via the upregulation of MYBS1 and CIPK15 under low-oxygen stress, thereby increasing carbohydrate availability and consequently enhancing AG tolerance [18] (Figure 2).

GWAS is an effective and popular strategy for dissecting complex traits, which can be able to capture complex trait variations by utilizing both evolutionary as well as historical recombination events at the population level [97]. In recent studies, 15 AG tolerance loci were detected through GWAS and one candidate gene, a DUF domain-containing protein (LOC_0s06g03520) was identified and found to be highly induced by AG [94]. Additionally, two candidate genes (LOC_Os03g31550 and LOC_Os12g31350) which encode *xanthine dehydrogenase 1* (OsXDH1) and *SSXT* family protein, respectively, have been identified using GWAS approach [96]. Nonetheless, the candidate genes currently isolated from rice seedlings for AG do not always overlap in the results of QTL mapping and GWAS. These may result from the different principle of two strategies. QTL mapping is based on linkage analysis of molecular markers using artificially constructed populations, while GWAS is performed based on the association analysis between single nucleotide polymorphisms (SNPs) and phenotypes and is able to uncover a large number of associated loci in natural populations. Although many QTLs/associated loci have been mapped using molecular marker assisted selection and GWAS, respectively; however, it is required for the identification of more AG tolerance genes in order to insightfully elucidate the molecular mechanisms underlying AG tolerance and provide major loci for breeding the AG tolerant varieties via molecular breeding.

## 6. Breeding Applications Using QTLs/Genes Underlying AG Tolerance

Introgression of the loci conferring submergence tolerance into rice mega varieties using marker-assisted selection has been extensively explored recently. The results showed that introgression has significantly enhanced submergence tolerance and did not exhibit negative impacts on rice development, yield, or grain quality [98,99,100,101].

Under field trials, the *qAG-9-2*-containing near-isogenic line (NIL-*AG1*), which carries a small Khao Hlan On (KHO) introgression in an elite cultivar (IR64) background, confers AG tolerance in the field and exhibits similar grain quality as IR64 [18]. The introgression of *AG1* into Ciherang-Sub1 significantly increased AG tolerance [101]. For rice direct seeding in the field, *AG1* and *AG2* introgression lines show no negative impact on the early seedling growth and exhibit yield stability under flooding condition [100], suggesting that the development of high-yielding varieties with submergence tolerance is feasible and imperative for the direct seeding practice.

In further studies, the more commonly detected QTLs and their candidate genes could provide promising perspectives for molecular genetic characterization of AG tolerance and for the rapid development of cultivars with enhanced submergence tolerance. The identified QTLs/genes are in favor of developing a more effective and efficient breeding strategy for direct seeding practice around the world [100,102].

## 7. Conclusions and Perspectives

The complex trait AG is influenced by many factors, including intrinsic genetic factors and seed quality factors, such as dormancy, maturity, and storage of nutrients, and environmental factors, such as light intensity, water oxygen content, pH, salinity, temperature, and soil physical state. Considering the major obstacle encountered in the direct seeding practice, poor seedling establishment under anerobic germination, it is critical for rice varieties to possess the ability to elongate the coleoptile faster to reach the water surface and escape the low O_2_ condition to ensure normal early growth [103]. Rice with faster coleoptile elongation can be applied in direct seeding and breeding improvement [104,105]. Compared with AG-tolerant varieties (LOES), AG-intolerant varieties (LOQS) severely inhibit growth and have lower survival rate under submergence; however, AG-intolerant varieties that have the characteristics of avoiding growth under submergence conditions also seem to be meaningful for short-term flooding in direct seeding applications in the paddy fields.

Here, we highlighted the important genes and regulators that underlie the essential mechanisms for AG tolerance. Among the stress-responsive pathways, the *CIPK15*-mediated O_2_-deficiency signaling regulatory pathway plays a key role in controlling sugar and energy production during AG. *CIPK15* regulates the *SnRK1A*-dependent sugar sensing, thereby regulating the abundance of αAmy and ADH for the metabolic shift and energy production to adapt to submergence conditions [15,40]. The upregulation of the key enzyme *αAmy 3* is effective in mobilizing starch to produce energy and shifts ATP production from aerobic to anaerobic respiration through signaling cascades and metabolic regulation under low O_2_ stress.

Recent progress allows us to better understand the stress-related regulatory mechanisms and the field trials. However, more extensive investigations are required to resolve global food security challenges in the future. It is necessary to highlight the significance of trait-to-gene-to-field approaches to ensure that appropriate coping strategies are adopted to enable staple food crops that are more tolerant to various abiotic stresses to maintain yield stability in the field [106,107].

## Figures and Tables

**Figure 1 plants-09-01363-f001:**
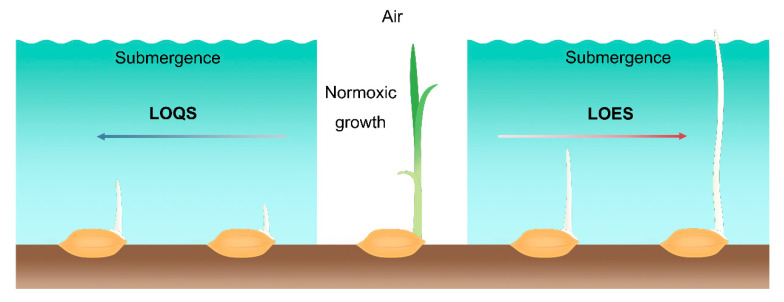
Low O_2_ escape strategy (LOES) and low O_2_ quiescence strategy (LOQS) strategy of rice germination under submergence conditions. Oxygen is one of the important factors that affects the germination and early seedling growth of rice. Under normoxic conditions, the surrounding oxygen ensures normal seed germination and early growth of the seedlings. Under the submergence condition, the formation of roots is inhibited in rice seedlings after germination. Sensitive cultivars adopt the quiescence strategy that inhibits germination and coleoptile elongation under low O_2_ stress, while tolerant varieties show phenotypes with faster coleoptile elongation capacity to reach the water surface for air exchange.

**Figure 2 plants-09-01363-f002:**
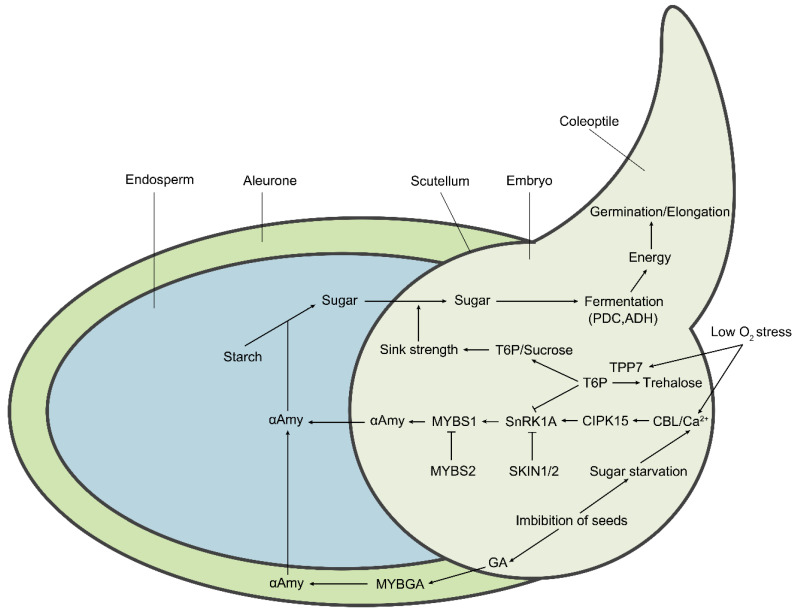
The molecular regulatory pathways and metabolic pathways of rice under hypoxic germination and early seedling growth. Sugar starvation caused by low O_2_ stress under the submergence condition is the key upstream signal affecting metabolic regulation pathways. The Ca^2+^ signal acts as a secondary messenger to mediate the downstream responses. The calcineurin B-like (CBL) proteins bind to Ca^2+^ and then interact with CIPK15, leading to the activation of its kinase activity. Subsequently, the activated CIPK15 physically interacts with SnRK1A, an upstream protein kinase of the transcription factor MYBS1, and activates its activity, consequently elevating the activity of αAmy for seed stored starch degradation. *SKIN1/2* and *MYBS2* negatively regulate SnRK1A and MYBS1, respectively, thereby repressing *αAmy* expression. Additionally, seed imbibition also results in the biosynthesis of gibberellin (GA) in the embryo. GA induces the expression of *MYBGA* in cereal aleurone cells, thereby upregulating the expression of the *αAmy* gene. Meanwhile, low O_2_-induced *OsTPP7* desuppresses the SnRK1A activity that is inhibited by trehalose 6-phosphate (T6P). OsTPP7 increases the sink strength of the embryo axis–coleoptile by the perception of low sugar availability by enhancing the conversion of T6P to trehalose and leads to a decrease in the T6P/sucrose ratio, thus enhancing starch mobilization for energy production to promote coleoptile elongation. In addition, low O_2_ results in the shift of aerobic respiration to anaerobic fermentation, therefore inducing the expression of essential components, including *PDC* and *ADH*.

**Table 1 plants-09-01363-t001:** Subfamilies of the rice αAmy genes.

Gene	Subfamily	Chromosome	Locus Name	CDS Coordinates (5′-3′)	Regulatory Function
RAmy1 A	RAmy1	2	LOC_Os02g52700	32243146-32245056	High temperature in developing seeds [58]
RAmy1 B		2	LOC_Os02g52710	32250180-32248279	Chemical inhibition [59]
RAmy1 C		1	LOC_Os01g25510	14459951-14461849	High temperature in developing seeds [58]
RAmy2 A	RAmy2	6	LOC_Os06g49970	30262778-30266915	Unknown
RAmy3 A	RAmy3	9	LOC_Os09g28400	17288993-17291295	High temperature in developing seeds [58]
RAmy3 B		9	LOC_Os09g28420	17296166-17305076	Chemical inhibition [59]
RAmy3 C		8	LOC_Os08g36900	23335165-23337151	Unknown
RAmy3 D		8	LOC_Os08g36910	23340676-23343533	High temperature in developing seeds [58] Sugar starvation [53,60] Calcium signaling [21]
RAmy3 E		4	LOC_Os04g33040	20006128-2000927	High temperature in developing seeds [58] Chemical inhibition [59]
RAmy3 F		1	LOC_Os01g51754	29760719-29770037	Unknown

**Table 2 plants-09-01363-t002:** The identified quantitative trait loci (QTLs)/candidate genes for anaerobic germination (AG) tolerance.

QTLs/Candidate Genes	Chromosome	Traits	Description & Reference
*qAG-1-2*	1	survival rate	[87]
*qAG-3-1*	3		
*qAG-7-2*	7		
*qAG-9-1, qAG-9-2*	9		
*qAG7.1*	7	survival rate	[89]
*qAG7*	7	survival rate	[90]
*qAG11*	11		
*qAG2.1*	2		
*qAG1a, qAG1b*	1	survival rate	[91]
*qAG8*	8		
*qAG7.1, qAG7.2, qAG7.3*	7	survival rate	[88]
*qAG3*	3		
*qSUR6–1*	6	survival rateseedling height	[92]
*qSH1–1*	1		
*OsTPP7*	9	coleoptile length	Enhancing germination and coleoptile elongation [18]
*HXK6*	1	coleoptile length	Encoded a hexokinase [93]
LOC_Os06g03520	6	coleoptile length	DUF domain containing protein [94]
*TIR1*	5	coleoptile length	F-Box auxin receptor protein [95] Control lateral root initiation in rice [95] Codes for an E3 ubiquitin ligase complex component [95] The ATP binding cassette transporter [95]
*AUX1*	1		
*COI1a*	1		
*ABC1-2*	2		
LOC_Os03g31550	3	survival rate and coleoptile	encodes for enzyme *xanthine dehydrogenase 1*(*OsXDH1*) [96] encodes for *SSXT* family protein [96]
LOC_Os12g31350	12		

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
