# Peer review of "The Molecular Regulatory Pathways and Metabolic Adaptation in the Seed Germination and Early Seedling Growth of Rice in Response to Low O2 Stress"

_plants, 2020, doi:10.3390/plants9101363_

Round 1
Reviewer 1 Report
see attached file

Author Response
Rice cultivars germinating and growing under hypoxic conditions potentially allow sowing in paddy fields, which is much more efficient then classical methods of rice cultivation relying on the planting of rice seedlings. Hence, there is much interest in identifying and breeding such cultivars. In this review, the authors link this trait to energy metabolism in seeds and coleoptiles relying on fermentation. Efficient regulation of fermentation to provide enough ATP, thus compensating for the poor efficiency of this pathway compared to respiration, is identified as a key process. However, this is a rather one-sided view and does not reflect the complexity of the adaptation to oxygen deprivation, and other factors is not given due attention. Among these is the avoidance of damage due to the accumulation of phytotoxic substances such as reduced iron (Fe+2), manganese (Mn+2), hydrogen sulphide (H2S), oxygen radicals, and the products of fermentation (ethanol). Moreover, the role of EXPA, EXPB, SUB1A and SNORKEL genes in mediating coleoptile growth at low oxygen, and avoidance of acidosis have to be mentioned. The role of phytohormones is also not considered. In these respects, the review falls short of previous reviews of the issue, e.g. by Miro and Ismail 2013 Front. Plant Sci. 4:269.
Reply: Thanks for the insightful comments! We have carefully considered your views and compared the references you provided. Just as your views, we did not mention some aspects in our paper so as to fail to fully reflect the complexity of the adaptation of rice to low oxygen stress. Therefore, we supplemented these factors you have mentioned in our manuscript accordingly: Add additional text in lines 49-52 involved the avoidance of damage due to the accumulation of phytotoxic substances, oxygen radicals, and the products of fermentation. Add additional text in lines 111-114 involved the role of EXPA, EXPB. Add additional text in lines 79-90 involved two adaptive strategies in established rice plants: snorkel 1 and snorkel 2 in LOES, and Sub1A in LOQS. Add additional text in lines 171-175 to mention the avoidance of acidosis. Add additional text in lines 256-261 involved phytohormones. Thanks!
Further aspects:
- 22 This review focusses on… (use simple present!)
Reply: We have changed “focused” to “focusses” in line 23.
- 24 Further, we highlight…
Reply: We have changed “highlighted” to “highlight” in line 25.
- 24 I object to the definition of hypoxia. It should be related to the critical oxygen pressure for respiration (COPR), which is the lowest oxygen partial pressure which allows unrestricted respiration (e.g. Wegner 2010 oxygen transport in waterlogged plants. In: S. Mancuso, S Shabala, waterlogging signalling and tolerance in plants, Springer). Under hypoxic conditions, some respiration can continue, though at a lower rate.
Reply: Thanks for the comment! We have added additional text involved definition of hypoxia related to the critical oxygen pressure for respiration (COPR). Please see lines 45-49.
- 56ff Use simple present!
Reply: We have changed “focused” to “focus” in line 68.
- 63 “AG tolerance”. Define abbreviations in the main text again.
Reply: We have defined abbreviations of “AG tolerance” in line 75.
- 170 CBL4 is induced by low oxygen, but its silencing downregulates ADH1. That doesn’t seem to make sense!
Reply: Thanks for the comment! We have deleted “and ADH1” in line 203.
- 248 Define GWAS
Reply: We have added additional text involved definition of GWAS in lines 285-287.
Figure 1: The figure is not very helpful in the present form. The LOES and LOQS strategy should be in the headings for the sketches. Arguably, LOQS implies avoidance of growth under flooding conditions, which can also a good alternative to paddy fields.
Reply: Thanks for the comments and suggestions! The form of Figure 1 has been modified and the headings for the sketches have revised. Please see line 101. In addition, we have added additional text involved LOQS of AG-intolerant varieties in lines 326-329.
Reviewer 2 Report
This review article aims to describe the important advances on the molecular mechanisms and metabolic adaptions under anaerobic germination (AG) and coleoptile elongation during rice seed germination. This article also introduces the candidate genes from QTL mapping and GWAS study for AG tolerance. This review manuscript which provided an important and valuable information about rice seed germination in response to hypoxia stress.
Below are some suggestions for improving the manuscript abundance.
- Add additional text involved the snorkel 1 and snorkel 2 in LOES, and Sub1A, ADH and SLR1 in LOQS.
- Discuss why the candidate genes from the results of QTL and GWAS almost do not overlap with the AG-related genes currently isolated from rice seedlings.
- Minor revise, line 32 (Oryza sativa should be changed to italics), line 262 (On change to on), line 123 (12 h light/12 h dark ?), line 124 (100 mM IAA, GA and ABA, the conc. is too high). The references # 1, 2, 8, 10, 12, 16, 17, 24, 27, 31, 36-38, 45, 47, 57, 69, 80-82, 90 and 91, the letter format of the title is wrong (capital letters should be changed to lowercase).
Author Response
This review article aims to describe the important advances on the molecular mechanisms and metabolic adaptions under anaerobic germination (AG) and coleoptile elongation during rice seed germination. This article also introduces the candidate genes from QTL mapping and GWAS study for AG tolerance. This review manuscript which provided an important and valuable information about rice seed germination in response to hypoxia stress.
Below are some suggestions for improving the manuscript abundance.
- Add additional text involved the snorkel 1 and snorkel 2 in LOES, and Sub1A, ADH and SLR1 in LOQS.
Reply: Thanks for the insightful comments and suggestions! We have added additional text involved two adaptive strategies in established rice plants: snorkel 1 and snorkel 2 in LOES, and Sub1A, ADH and SLR1 in LOQS. Thanks! Please see lines 79-90.
- Discuss why the candidate genes from the results of QTL and GWAS almost do not overlap with the AG-related genes currently isolated from rice seedlings.
Reply: We have added sentences to discuss the issue relate to QTL and GWAS. Please see lines 291-297.
- Minor revise, line 32 (Oryza sativa should be changed to italics), line 262 (On change to on), line 123 (12 h light/12 h dark ?), line 124 (100 mM IAA, GA and ABA, the conc. is too high). The references # 1, 2, 8, 10, 12, 16, 17, 24, 27, 31, 36-38, 45, 47, 57, 69, 80-82, 90 and 91, the letter format of the title is wrong (capital letters should be changed to lowercase).
Reply: Thanks for reading the manuscript carefully, we have changed “Oryza sativa” to italics in line 32. The name of rice variety “Khao Hlan On” is correct and we add its abbreviations “KHO” in line 307. In addition, other revise suggestions are not applicable to our manuscript.
Reviewer 3 Report
The paper “The molecular regulatory pathways and metabolic adaptation in the seed germination and early seedling growth of rice in response to low O2 stress” is a review article. It is presenting the molecular metabolic background of the universal plants adaptation ability phenomenon using the example of cultivated grass Rice (Oryza sativa L.). The authors have presented a review which is a result of the analysis of 94 newest papers. The review is focused on concerning the groups of individuals of cultivated grass species Rice (Oryza sativa L.) undergoing low oxygen (O2) stress caused by flooding/submergence.
The understanding of the mechanisms underlying the traits variety of plant individual’s responses to environmental circumstances is of high importance also for the ecosystem development understanding. Scientific achievements in this respect are of high knowledge development and application value. Agricultural plant applications depend, to some extent, on the range on the natural plants ability to new and/or difficult environmental conditions. This concept is in my opinion not explored enough in this paper. All the agricultural applications are possible only due to natural phenotypic plasticity and astonishing adaptive potential in plants (also the cultivated ones).
The submergence environmental conditions provide limited O2 and carbon dioxide (CO2) availability in the submerged plant parts. The plants growing in such conditions have to posses the complex trait of anaerobic germination (AG) ability. This trait is influenced by intrinsic genetic factors, seed quality factors (dormancy, storage of nutrients) and environmental factors (light, temperature, salinity, pH, oxygen content in the water and soil physical parameters).
The submergence environmental conditions requires the seedlings/plants establishment under anaerobic germination. In this review two distinct adaptive survival strategies of response to submergence stress in plants: i.) the low O2 escape strategy (LOES); ii.) the low O2 quiescence strategy (LOQS).
Author Response
The paper “The molecular regulatory pathways and metabolic adaptation in the seed germination and early seedling growth of rice in response to low O2 stress” is a review article. It is presenting the molecular metabolic background of the universal plants adaptation ability phenomenon using the example of cultivated grass Rice (Oryza sativa L.). The authors have presented a review which is a result of the analysis of 94 newest papers. The review is focused on concerning the groups of individuals of cultivated grass species Rice (Oryza sativa L.) undergoing low oxygen (O2) stress caused by flooding/submergence.
The understanding of the mechanisms underlying the traits variety of plant individual’s responses to environmental circumstances is of high importance also for the ecosystem development understanding. Scientific achievements in this respect are of high knowledge development and application value. Agricultural plant applications depend, to some extent, on the range on the natural plants ability to new and/or difficult environmental conditions. This concept is in my opinion not explored enough in this paper. All the agricultural applications are possible only due to natural phenotypic plasticity and astonishing adaptive potential in plants (also the cultivated ones).
The submergence environmental conditions provide limited O2 and carbon dioxide (CO2) availability in the submerged plant parts. The plants growing in such conditions have to posses the complex trait of anaerobic germination (AG) ability. This trait is influenced by intrinsic genetic factors, seed quality factors (dormancy, storage of nutrients) and environmental factors (light, temperature, salinity, pH, oxygen content in the water and soil physical parameters).
The submergence environmental conditions requires the seedlings/plants establishment under anaerobic germination. In this review two distinct adaptive survival strategies of response to submergence stress in plants: i.) the low O2 escape strategy (LOES); ii.) the low O2 quiescence strategy (LOQS).
Reply: Thanks for the insightful comments! The concept of “agricultural plant applications depend on the range on the natural plants ability to new and/or difficult environmental conditions” is indeed to be considered and explored. Therefore, we have added additional text to highlight and explore the issue. Please see lines 52-57. Thanks!
In addition, some minor revisions were made with change tracking pattern.
We have added 13 references (# 10-12, 30-34, 37, 38, 85, 86, 97).
line 487 (deleted “42.”), and line 599 (deleted “90.”).
“alcohol dehydrogenase (ADH)” was changed to “ADH” in line 188.